# Acid-Sensing Ion Channels and Mechanosensation

**DOI:** 10.3390/ijms22094810

**Published:** 2021-05-01

**Authors:** Nina Ruan, Jacob Tribble, Andrew M. Peterson, Qian Jiang, John Q. Wang, Xiang-Ping Chu

**Affiliations:** Department of Biomedical Sciences, School of Medicine, University of Missouri, Kansas City, MO 64108, USA; nrgrf@umkc.edu (N.R.); jttbkh@umkc.edu (J.T.); amp8m6@umkc.edu (A.M.P.); jiangqi@umkc.edu (Q.J.); wangjq@umkc.edu (J.Q.W.)

**Keywords:** acid-sensing ion channels, mechanosensation, neurodegenerative diseases, nociception

## Abstract

Acid-sensing ion channels (ASICs) are mainly proton-gated cation channels that are activated by pH drops and nonproton ligands. They are part of the degenerin/epithelial sodium channel superfamily due to their sodium permeability. Predominantly expressed in the central nervous system, ASICs are involved in synaptic plasticity, learning/memory, and fear conditioning. These channels have also been implicated in multiple disease conditions, including ischemic brain injury, multiple sclerosis, Alzheimer’s disease, and drug addiction. Recent research has illustrated the involvement of ASICs in mechanosensation. Mechanosensation is a form of signal transduction in which mechanical forces are converted into neuronal signals. Specific mechanosensitive functions have been elucidated in functional ASIC1a, ASIC1b, ASIC2a, and ASIC3. The implications of mechanosensation in ASICs indicate their subsequent involvement in functions such as maintaining blood pressure, modulating the gastrointestinal function, and bladder micturition, and contributing to nociception. The underlying mechanism of ASIC mechanosensation is the tether-gate model, which uses a gating-spring mechanism to activate ASIC responses. Further understanding of the mechanism of ASICs will help in treatments for ASIC-related pathologies. Along with the well-known chemosensitive functions of ASICs, emerging evidence has revealed that mechanosensitive functions of ASICs are important for maintaining homeostasis and contribute to various disease conditions.

## 1. Introduction

Acid-sensing ion channels (ASICs) are mainly proton-gated cation channels [1], which can be activated by a drop in extracellular pH below 7.0 and triggered by nonproton ligands during physiological pH levels [2]. There are currently at least six identified ASIC isoforms (ASIC1a, 1b, 2a, 2b, 3, and 4) encoded by four genes (*Accn1*, *Accn2*, *Accn3,* and *Accn4*) [3,4]. Activation of ASICs mostly triggers Na^+^ influx. As such, ASICs belong to the degenerin/epithelial sodium channel (DEG/ENaC) superfamily of ion channels. These channels are made up of 500–560 amino acids [3,5]. The general structure of an ASIC includes a homotrimeric or heterotrimeric proton-gated channel [6,7]. Each of the individual subunits is shaped like a “clenched fist” with six domains: wrist, finger, β-ball, thumb, knuckle, and palm domains in the extracellular loop [3,5]. The overall structure consists of an intracellular N terminus and an intracellular C terminus with two transmembrane domains (TMs) that are voltage-independent and help recognize extracellular ligands to regulate proton-gated currents [8]. In addition to the permeability to Na^+^, activation of ASICs also exhibits calcium permeability in certain subunits, such as ASIC1a, a mechanism important for the regulation of presynaptic neurotransmitter release and ultimately for regulatory functions, such as synaptic plasticity, learning, and memory [9,10].

ASIC1a is enriched in neurons and is distributed in various subcellular regions, including dendrites, dendritic spines, axons, neuron cell bodies, and intracellular organelles, such as the mitochondria [11,12]. This channel is an important mediator of acid-activated responses as well as acidosis-induced physiological changes in the central nervous system (CNS) and peripheral nervous system (PNS) [9,13]. Recent research has found that ASIC1a is linked to fear-related behaviors [14,15]. This linkage has been suggested by a high expression of ASIC1a in the fear-related forebrain regions, such as the amygdala, dorsal striatum, and nucleus accumbens [14]. Results from numerous studies have shown decreased fear conditioning in ASIC1a knockout (KO) mice (ASIC1a^−/−^) [14,15,16]. Furthermore, ASIC1a has been found to play a critical role in synaptic plasticity [17,18]. For example, long-term potentiation and spatial memory were decreased in ASIC1a^−/−^ mice, while long-term depression was increased in ASIC1a KO mice [17,18]. The expansive nature of ASIC1a gives it a wide range of neuromodulation functions [19]. For instance, activation of ASIC1a facilitated *N*-methyl-*D*-aspartate (NMDA) receptor function [20]. Apart from being expressed in the nervous system, ASIC1a is found in peripheral tissues, including the vasculature and intestines [4]. The roles these channels play in the vasculature include vasoconstriction, vascular hypertrophy, and vascular remodeling [21,22]. Recent studies have found that these effects on vascular reactivity are found within the pulmonary vasculature but not within the mesenteric vasculature [22].

ASIC1b is found primarily in peripheral sensory neurons [23]. Studies have shown that ASIC1b containing channels are heterogeneous and may form heterotrimeric channels with other ASIC subtypes such as ASIC3 or ASIC1a [23]. Functional ASIC1b plays a role in peripheral nociception and pain [24,25].

Like ASIC1, ASIC2 has two variants: ASIC2a and ASIC2b. ASIC2 is widely expressed throughout the brain, including the hippocampus, cortex, amygdala, and olfactory bulb [26,27]. ASIC2 is also associated with other ASIC subunits, such as ASIC1a or ASIC3, to form heterotrimers [3,5]. For example, ASIC2/1a is a dominant subunit expressed in the cortex, and deletion of ASIC2 leads to decreased membrane trafficking of ASIC1a to the cell surface [28]. Consistent with this idea, both ASIC2 and ASIC1a have been implicated in neuroprotection during disease states [29,30,31,32,33,34]. ASIC2 has also been implicated in baroreceptor activities related to the cardiovascular system [35]. The ASIC2 that regulates baroreceptor functions is located inside the nodose ganglia [35]. In mice with a KO of the ASIC2 gene (ASIC2^−/−^), there is a decreased baroreceptor function [35]. Besides the pulmonary and cardiac vasculature, immunolabeling studies reveal that ASIC2 plays a role in renal vasculature [36]. The function of ASIC2 in the kidney is to help with the myogenic regulation of renal blood flow [37]. Although there are many functions of ASIC2 that need to be clarified, one of the functions of this channel is to regulate blood flow. ASIC2a also exerts its protection against acid-induced rat articular chondrocyte apoptosis through regulating ASIC1a expression and the intracellular Ca^2+^ levels and, at least in part, suppressing p38 and extracellular signal-regulated kinase 1/2 mitogen-activated protein kinase signaling pathways [38]. Different from ASIC2a [39], ASIC2b cannot form a functional channel by itself [3,19]. We do know, however, that ASIC2b has been found to modulate the properties of other ASICs, such as ASIC1a, by forming heterotrimeric channels [40,41]. For example, ASIC2b forms a functional channel with ASIC1a. Such heterotrimeric channel shows calcium permeability and contributes to ischemic brain injury [40].

ASIC3 is mainly located in the peripheral dorsal root ganglion (DRG) neurons [4,42]. Other locations of ASIC3 in the PNS include the spiral ganglia, nodose ganglia, trigeminal ganglia, and neurons in the bladder [43]. In addition to forming functional channels with other ASIC subunits, such as ASIC1a [44], ASIC3 is also associated with the P2X3 ion channel [45]. Although ASIC3 largely contributes to pain modulation [46,47,48], ASIC3 also plays a role in the bladder. In mice lacking the ASIC3 gene (ASIC3^−/−^), problems were found with regards to micturition, including voiding and oliguria [49].

ASIC4 is primary located in the pituitary gland [50]. Other locations of ASIC4 include the olfactory bulb, hippocampus, caudate putamen, amygdala, cerebral cortices, thalamus, brainstem, spinal cord, and the preoptic area [51,52]. Like ASIC2b, ASIC4 does not form a functional channel by itself and is not activated by protons [53]. Currently, the exact physiological stimulus that activates ASIC4 is unknown [53,54]. Because of the unique characteristics of ASIC4, there are a lot of unknowns when it comes to the expression and function of this channel. One other location where ASIC4 is seen and is clinically relevant is osteoblasts. ASIC4, along with ASIC2 and ASIC3, are highly expressed during osteoblastogenesis in an acid environment [55]. The function of this event is yet to be elucidated. Research has looked at correlations between ASIC1a and ASIC4 [52]. Recent studies have found that ASIC4 may modulate the innate fear response via the predator odor and anxious state [52]. The mechanism underlying the ASIC4-mediated modulation of fear responses may involve the role of ASIC4 in regulating ASIC1a in the brain [52].

## 2. ASIC-Associated Pathologies

In addition to its physiological roles, dysfunctional ASIC1a is largely linked to disease states [9,19,54]. ASIC1a is extensively involved in neurological and psychological diseases, such as ischemic brain injury [28,33], traumatic brain and spinal cord injury [29,30], Parkinson’s disease (PD) [31], Alzheimer’s disease (AD) [32], experimental autoimmune encephalomyelitis (EAE) [56], multiple sclerosis (MS) [57], seizure disorders [58], pain [59,60], and drug addiction [61,62,63,64,65].

ASICs are implicated in drug addiction. For example, ASIC1a^−/−^ and ASIC2^−/−^ mice were both found to cause increases in the α-amino-3-hydroxy-5-methyl-4-isoxazolepropionic acid (AMPA):NMDA receptor ratio and dendritic spine density during cocaine addiction [64]. This suggests a protective role of ASIC1a and ASIC2 in drug addition by inhibiting cocaine-induced plasticity [62,65]. However, according to a study with overexpression of ASIC1a in the nucleus accumbens, ASIC1a plays a role in the underlying extinction and cocaine-seeking behavior, underscoring the complex roles of ASIC1a in synaptic plasticity related to drug addiction [61].

Blockade of ASIC1a channels in the proximal tubule attenuated Ca^2+^ influx in instances of renal ischemic reperfusion and led to decreased levels of human proximal tubular cell apoptosis, indicating that ASIC1a contributes to reperfusion-induced injuries in the kidney [66]. In the brain, our studies have shown that disruption of either ASIC1 or ASIC2 genes exerted neuroprotection against ischemic brain injury [28,67]. Deletion of ASIC2 significantly decreased the trafficking of ASIC1a to the cell membrane and reduced the infarct volume of the brain in an experimental ischemic stroke model [28]. This enforces the functional role of ASIC1a and ASIC2 in ischemic brain injury. Along the lines of protective mechanisms in the brain, ASIC2 has also been shown to protect humans from pulmonary hypertension by increasing the vasoreactivity of the pulmonary vasculature [34]. ASIC1a acidification additionally showed the capacity of recruiting the receptor-interacting serine/threonine-protein kinase 1 (RIPK1) during the development of ischemic neuronal injury, which ultimately leads to neuronal cell death [68]. Thus, when targeting 20 N-terminus residues of ASIC1a in an ischemic mouse model, it subsequently demonstrated therapeutic potential by preventing RIPK1 activation for possible protection against neuronal cell death [68]. However, it was also found that the ASIC1a N-terminus spontaneously bound the N-ethymaleimide-sensitive fusion ATPase (NSF) during acidic conditions. As an important molecule for synaptic vesicle fusion, NSF is worth further investigation for its roles in the auto-inhibition of ASIC1a without interfering with ASIC1a’s desirable physiological functions [69]. These studies also introduce the therapeutic potential of ASIC1a-blocking monoclonal antibodies, such as ASC06-IgG1 [70,71]. β-estradiol has also been reported for stroke treatment via a similar mechanism involving the downregulation of ASIC1a [33]. Moreover, inhibition of the neuropeptides-ASIC1a interaction reveals neuroprotection during ischemic brain injury [72,73].

ASICs are involved in MS [57]. Acidotoxicity enhances the influx of Ca^2+^ and Na^+^ through ASIC1a, which ultimately causes neuronal degeneration and inflammatory reactions in MS [57,74,75]. The finding that the expression of ASIC1a in axons and oligodendrocytes was increased in MS patients with increased axonal injury reinforces the notion that ASIC1a is an essential player in MS [74]. ASIC2^−/−^ mice were also observed to have a significantly decreased clinical score for MS, which identifies ASIC2 as a potential player implicated in MS. Similar to ASIC1, ASIC2 plays a detrimental role in MS and exacerbates axon degeneration [76]. Consistent with the role of ASIC1a and ASIC2 in promoting MS, ASIC blockers, such as amiloride, have neuroprotective properties [77,78]. However, the majority of studies on ASICs and MS have been conducted either in vitro or in animal models. Studies on humans are warranted to evaluate the clinical implications of ASICs in MS [79].

It has been suggested that ASICs are involved in forming the acidic environment in the pathologic AD brain, although the exact underlying mechanism remains elusive [32]. For instance, upregulation of ASIC1a led to a dramatic increase in intracellular Ca^2+^, which helps maintain the acidic brain environment for the ultimate degeneration of microglial cells [80,81]. Additionally, in the presence of Aβ and an agonist for group I metabotropic glutamate (mGlu) receptors, ASIC1a triggered an increase in intrinsic excitability of hippocampal neurons, indicating a functional coupling between ASIC1a and group I mGlu receptors in the remodeling of synaptic transmission critical for AD [32]. Interestingly, the common drug for clinical treatment of mild to moderate AD patients, memantine, has been shown to inhibit ASIC1a along with its well-known mechanism of inhibition of NMDA receptors [82]. With the further exploration of the ASIC-dependent mechanisms underlying AD, more therapeutic agents for AD by targeting ASICs are expected to be developed in the future.

ASIC1b is linked to pain sensation [23,24,25]. Transient and long-lasting mechanical hyperalgesia in ASIC1b wild-type (ASIC1b^+/+^) mice has been shown to last much longer than in ASIC1b^−/−^ mice, though future research is needed to delineate its mechanism further [23]. In a study carried out by Lee et al. (2018), antihyperalgesic medication at higher doses significantly reduced ASIC1b activity in rodents [25]. This reveals a correlation between the peripheral nociception and ASIC1b.

Like ASIC1b, ASIC3 plays a critical role in the pain pathway [47,48]. Current studies have shown that the hyperalgesic response towards muscle inflammation was eliminated in ASIC3 KO mice [47]. The mechanism underlying the role of ASIC3 has been suggested. Namely, muscle inflammation causes a local acidosis in the affected area. This acidosis triggers the proton sensing capabilities and activation of ASIC3, which elicits a pain signal [47]. Multiple ion channels have been reported to influence the nociception of ASIC3. One receptor, in particular, the proteinase-activated receptor 2, causes systemic sensitization of ASIC3 and, in turn, increases the pain response [47]. Another location of interest where ASIC3 populates is the gastrointestinal (GI) tract [4,48]. In the GI tract, ASIC3 participates in the inflammatory response towards gastric acid secretion under conditions such as gastritis or peptic ulcers [48].

There is an expanding avenue of research pertaining to pharmacological profiles of ASIC blockers in treating ASIC-associated pathologies [3,5,9]. As shown in Figure 1, spider-venom peptide psalmotoxin-**1** (PcTx1) is shown to be an important inhibitor of ASIC1a by binding the thumb α-helix 5 component of ASICs in rodent models [83,84]. It works in human ASIC1a and thus becomes an important analgesic and ischemic stroke therapy [85]. Another spider venom peptide known as Hm3a has recently been identified to inhibit ASIC1a and ASIC1b with a similar half amount of excitation concentration of PcTx1 and higher levels of stability across 48 h (Figure 1). This peptide alleviates symptoms of potentially both MS and strokes [86]. Both diminazene and mambalgin-1 are also discovered to block the ASIC1a receptor, which is yet another avenue to inhibit acidosis in neuroinflammation [87,88]. APETx2, a sea anemones peptide and a selective ASIC3 blocker, have exhibited inhibition of transient ASIC3 current [89]. ASIC3 blockade significantly reduces fibromyalgia pain in mice [90]. A-317567 is a small molecule, non-amiloride ASIC blocker [91], which potently blocks ASICs, especially in the DRG [91]. Evidence shows that A-317567 is implicated in the treatment of many disorders, including chronic pain and irritable bladder conditions [92].

## 3. ASICs in Mechanosensation

Mechanosensation is an integral part of ion channels and a form of signal transduction in which mechanical forces are converted into neuronal signals [93]. The electrical signal that is created from mechanosensitive ion channels will then help mediate numerous bodily functions, including hearing, balance, proprioception, volume regulation of erythrocytes, nociception, vascular function, and touch [93]. Recent research in the field of ion channel-related mechanosensitive functions has further elucidated the importance and widespread functions of mechanosensation in ion channels. The functions of mechanosensation in ion channels seem to be far more substantial than simply the sensory aspect that has been elucidated for some time. The basic structure of mechanosensitive ion channels, in general, is a transmembrane protein with a mechanical gate requiring a stimulus to activate [94]. The neural circuits surrounding mechanosensation are not well understood, but they help process information to cause changes in behavior and maintain physiological homeostasis [94]. In regard to mechanosensation related specifically to ASICs, there has also been an exponential amount of research that has been published as of late, which has opened up many avenues for the future [4]. However, there are still many unknowns surrounding ASICs related mechanosensation, such as the exact physiological response that stimulates these channels. This has facilitated the research regarding ASICs and mechanosensation to flourish.

How ASICs play a role in mechanosensation remains unclear. A prevailing theory involves the mechanotransduction complex in *Caenorhabditis elegans*. The complex that senses gentle touch is composed of DEG/ENaC proteins, such as MEC-4 and MEC-10, which are connected to touch receptor neurons [95] (Figure 2). The complex is additionally supported by accessory subunits, such as MEC-2, MEC-6 and UNC-24, which show to have only a partial loss of touch sensitivity when deleted as they are not associated tightly with the MEC-4 and MEC-10 complex [95]. DEG/ENaC channel proteins and accessory proteins are all essential for mechanoreceptor activity because the mutation of *Mec2*, *Mec4*, and *Mec6* genes eliminated mechanoreceptor currents [96]. In addition, between the extracellular matrix (ECM) and the cytoskeleton of the organism, there are ECM-linker proteins, such as MEC-1 and MEC-9, as well as intracellular linker proteins, such as stomatin (STOM). They convey signals from the extracellular matrix to the cytoskeleton [97,98,99]. In a tether model, MEC-4/MEC-10 act as a gating-spring mechanism in mechanosensation [96,100]. When compared to nematodes, mammals are specifically characterized by STOM proteins, such as STOM-like 1 (STOML1) and STOM-like 3 (STOML3), as opposed to nematode accessory subunits [101] (Figure 2). STOM is a protein that is likely attached to the C-terminus of MEC-4/MEC-10 to connect with the TM1 of ASIC3 to suppress the ion channel [102]. Thus, in mice lacking STOM, stimulation of mechanosensation in D-hair receptors on the skin was reduced [103]. Additionally, ECM-linker proteins may otherwise cause extracellular tension and trigger a conformational change to activate the kindlin-integrin-RhoA pathway, which further stimulates mechanotransduction [104].

Another theory regarding the mechanism of mechanotransduction in ion channels involves PIEZO proteins and their roles in the “bilayer model” [105]. Some of the mechanosensitive functions that this channel is thought to carry out include proprioception, sensing light touch, and sensing stretch in organs [105]. This “bilayer model” theory has not been fully elucidated, and it is only linked to ion channels in general, not specifically to ASICs [104]. The bilayer model of mechanosensation is relatively simple as the mechanical stimulus directly modulates the gating of the ion channel [106]. This stimulus differs from tissue to tissue. For example, in blood vessels, the laminar and oscillatory shear stress of blood stimulates the ion channel, while in peripheral sensory neurons, external physical forces stimulate the ion channel [106]. The bilayer model has been well known for some time. However, the role that PIEZO proteins play in this theory is novel. PIEZO proteins are large integral membrane proteins with 24–40 TMs that are implicated in converting the mechanical force into biological signals, specifically in mammalian cells [107]. Other functions of PIEZO proteins in humans include their promotion of various cellular developmental events, including cellular migration, elongation, and proliferation [107]. Of the PIEZO family members, Piezo1 and Piezo2 have specifically been found to function in mechanosensation [108,109]. Evidence proving the involvement of Piezo1 and Piezo2 proteins in mechanosensation has been found in studies with mice [109]. In these studies, mice with disrupted Piezo1 and Piezo2 genes showed a lower level of mechanically activated cation channel activities [109]. The exact mechanism underlying the role of Piezo in mechanosensation remains unknown. However, studies from localization and fluorescence imaging of Piezo genes support their roles in mechanosensation [108].

ASIC1 channels are located widely in the visceral sensory ganglia [110]. In ASIC1^−/−^ mice, there is also a subsequent increase in mechanosensitivity in the esophageal and colonic afferent mechanotransduction, indicating the importance of ASIC1a in visceral mechanosensation [111]. ASIC1 also has an important visceral mechanosensation process in urothelium and bladder compliance sensation [92,112]. However, loss of the ASIC1 did not appear to affect any cutaneous mechanoreceptors [113]. Further, ASIC1 channels are found in human cutaneous Pacinian corpuscles and may serve as rapidly adapting low-threshold mechanoreceptors, suggesting specific roles of ASIC1 proteins in human mechanotransduction [114]. ASIC1 is shown to have an effect on primary hyperalgesia during inflammation, which is a local response in the area of injury [115]. On the other hand, ASIC1 does not particularly affect secondary hyperalgesia, which is a response outside the area of injury [115]. ASIC1a is also involved in the pain pathway. For example, blocking ASIC1a by PcTx1 results in the activation of the endogenous enkephalin pathway [59]. The results suggest that ASIC1a channel is an important molecular target for treating both acute and neuropathic pain and that PcTx1 itself could be a potential analgesic drug working upstream of the opiate receptors. Recently, ASIC1 has been linked to migraine [116]. For example, intravenous injection of amiloride and mambalgin-1 both exert long-lasting anti-allodynic effects against acute and chronic cutaneous allodynia in the isosorbide dinitrate-induced migraine model, suggesting the involvement of peripheral ASIC1 channels in migraine cutaneous allodynia as well as in its chronification. The results shield light on the therapeutic potential of ASIC1 inhibitors as both an acute and prophylactic treatment for migraine [116]. Current research also demonstrates a mechanosensory role of ASIC1 in peripheral vasoconstriction and vascular remodeling [117]. More specifically, ASIC1 participates in the modulation of mechanosensation through the PNS. In ASIC1^−/−^ mice, the activity of mechanoreceptors on visceral afferent nerves was enhanced, indicating that ASIC1 may, in some circumstances, decrease mechanosensation [118]. The presence of ASIC1 in other peripheral tissues includes arteries, bone marrow, intestine, tongue, and bladder, indicating possible involvement in mechanosensation [118,119,120]. Further, ASIC1b has been tied to mechanosensation due to its apparent involvement in the pain sensation [24,25]. When higher doses of antihyperalgesic agents were given in rodents, the ASIC1b channel activity was decreased [24], although exact mechanisms underlying mechanosensitive functions of ASIC1b are unclear.

ASIC2 is linked to mechanosensation in the PNS [121]. The specific neurons that have recently been implicated in nociception and mechanosensation include the Isolectin B4-binding DRG neurons [122]. ASIC2 is also involved in mechanosensation in the autonomic nervous system via the nodose ganglia [123]. The autonomic regulation by ASIC2 is vital, as ASIC2 regulates cardiac afferents to control blood pressure [124]. This function that ASIC2 has in regard to controlling blood pressure showcases the baroreceptive functions of ASIC2 [125]. For instance, in ASIC2 null mice, impaired baroreceptor reflex manifestations were shown, such as increased blood pressure and exaggerated sympathetic response, demonstrating the importance of ASIC2 in baroreception [35]. The underly mechanism of ASIC2 in the regulation of mechanosensitive properties might be due to the membrane trafficking of ASIC2 proteins [126], which needs to be examined by further study. Another area where ASIC2 acts to regulate mechanoreceptors is the skin [121]. Although the results are mixed, ASIC2 proteins are present in Meissner, Merkel, penicillate, reticular, lanceolate, and hair follicle palisades in the rat skin [110]. With regard to mechanisms, studies have shown functional connections between ASIC2 and the tether model of mechanosensation. For instance, STOML3, a protein involved with the tether model, inhibited the acid-induced current of ASIC2 [127]. STOML3 KO mice displayed a nearly 40% reduction in mechanoreceptor sensitivity to mechanical stimulus [127].

ASIC3 has been a heavily studied ion channel due to its wide distribution in the PNS [128] and connections to mechanosensation [4], especially ASIC3 involved in pain modulation. Although the data from different pain models regarding the ASIC3 involvement are mixed, ASIC3 largely contributes to pain modulation (see reviews) [129,130,131,132,133]. In the DRG, ASIC3 transduces mechanosensation through ECM-induced neuron stretching instead of direct neuron indentation [134]. Different from ASIC1 in hyperalgesia, ASIC3 contributes to the development of secondary but not primary hyperalgesia [115]. In the colon, ASIC3 is present as the most abundant subtype of mechanosensory channels among the ASIC family and plays a critical role in visceral pain [110]. Given the abundant expression and robust roles of ASIC3 in the colon, ASIC3 is considered to be a potential target for developing pharmacotherapies for visceral colonic pain [48]. Another location where ASIC3 exhibits mechanosensitive properties is in sensory nerves responsible for skeletal muscle [135]. ASIC3 has been activated during muscle ischemia. This activation helps induce the reflex baroreceptor response, which helps vasodilate the arteries that supply these skeletal muscles [135]. ASIC3 is also critical for maintaining proprioception in mice [130]. Dysfunction of this process may lead to neurodevelopmental disorders with social behavioral disorders [42]. Along with proprioception, human ASIC3 is very sensitive to pH drops, indicating that human ASIC3 might actively modulate nociception [136]. In studies conducted in ASIC3^−/−^ mice, a marked reduction in acid-induced pain was noticed, along with a decrease in the ability to prime nociceptors [136]. Moreover, ASIC3^−/−^ mice were unable to develop chronic muscle pain, establishing a clear correlation between ASIC3 and nociception [137]. In mouse DRG neurons, recent experiments have shown that ASIC3 expresses the dual function of mechanosensation and acid-sensation [138]. The mechanosensing aspect of the nerves is what gives the proprioception and nociception function [4]. This proprioceptive and nociceptive function is expressed through ASIC3 in free nerve endings of the skin and many types of cutaneous nerves projecting to mechanical sensory structures, including lanceolate fibers, Meissner corpuscles, and Merkel cells [110]. ASIC3 is also heavily implicated in bladder physiology by providing sensory signaling during the filling of the bladder [49]. In addition, ASIC3 is involved in pain sensation caused by inflammation in the bladder [112]. Specifically, the ASIC3-mediated mechanosensation was upregulated in the urothelium and suburothelial nerve plexus of the bladder during cystitis [112]. Lastly, recent studies concerning lampreys have linked ASIC3 to the mechanosensation of cerebrospinal fluid (CSF)-contacting neurons in the hypothalamus [139]. The mechanosensation of these neurons is induced by fluid movement along the walls of the third ventricle, which represents a regulatory feedback mechanism in lampreys to protect the CNS from changes in both pH and motion [139]. The mechanosensory function of ASIC3 has also been attributed to the aforementioned tether model, as has been seen in the DRG proprioceptors [4]. Further connections between the tether model and ASIC3 may exist since ASIC3-mediated currents were inhibited by STOML3 [127], similar to the observations with ASIC2a.

## 4. Perspective for Future Studies

One aspect of ASICs in mechanosensation is that it requires further testing on mammals. Many breakthroughs in the field of mechanosensation in relation to ASICs were made in species of non-mammalian origin. For example, the prevailing theory underlying the mechanosensation mechanism in ASICs, the tether model, was found in nematodes [95]. Most of the studies surrounding mammalian mechanosensation in ASICs have been conducted in mice. Having a more diverse set of mammals to gather data would facilitate elucidation of the unknowns behind ASIC mechanosensation. Further efforts are also needed to understand the mechanisms underlying the role of ASICs in mechanosensation. Although there is a leading theory concerning ASIC and mechanosensation, there are still many unknowns regarding the details and accuracy of this model [102]. Studies looking at these mechanisms in detail will advance our knowledge, especially the intracellular signaling pathway linking ASICs to mechanosensation. Another task of further studies is the attempt to explore ASIC4. ASIC4 is by far the least understood ASIC subtype [53,54]. The physiological stimulus that activates ASIC4 is relatively unknown, and neither is its function and expression. Further investigations of this channel are warranted to explore whether ASIC4 plays a meaningful role in mechanosensation. Finally, more studies need to be carried out to find additional agents that are useful for treating ASIC-related pathologies, including MS, strokes, and others. While we are currently looking more into spider venom peptides, e.g., PcTx1 and Hm3a, we need to illuminate underlying mechanisms in order to enhance the efficacy of these agents and prevent possible side effects in the therapy of ASIC-associated disorders [84,86].

## 5. Conclusions

In conclusion, the mechanosensitive function of ASICs is a rapidly growing field of research. Some of the most pertinent roles of ASICs in mechanosensation include the modulation of nociception, bladder activity, blood pressure, and activity of the GI tract, in addition to numerous other responses (see Table 1). Both ASIC1a and ASIC1b are present in the PNS. ASIC1a modulates mechanosensation by inhibiting the function of visceral afferent nerves, while ASIC1b regulates the pain pathway by increasing nociception. ASIC1a and ASIC1b can work jointly to modulate vasoconstriction of blood vessels, gastric emptying, and bladder compliance. ASIC2a is expressed in the nodose ganglion and controls the mechanosensitive functioning in the autonomic nervous system. The regulation through autonomics is attributed to the modulation of cardiac afferents. The mechanosensative function of ASIC3 is focused on pain response. Other functions include proprioception at cutaneous nerves and sensory signaling during the filling of the bladder. In the future, we need to clarify further the mechanisms underlying the role of ASICs in mechanosensation and to explore ASIC4 and its contributions to mechanosensation. These research endeavors will promote the development of a new generation of pharmacotherapies by targeting ASICs for the treatment of ASIC-associated disorders.

## Figures and Tables

**Figure 1 ijms-22-04810-f001:**
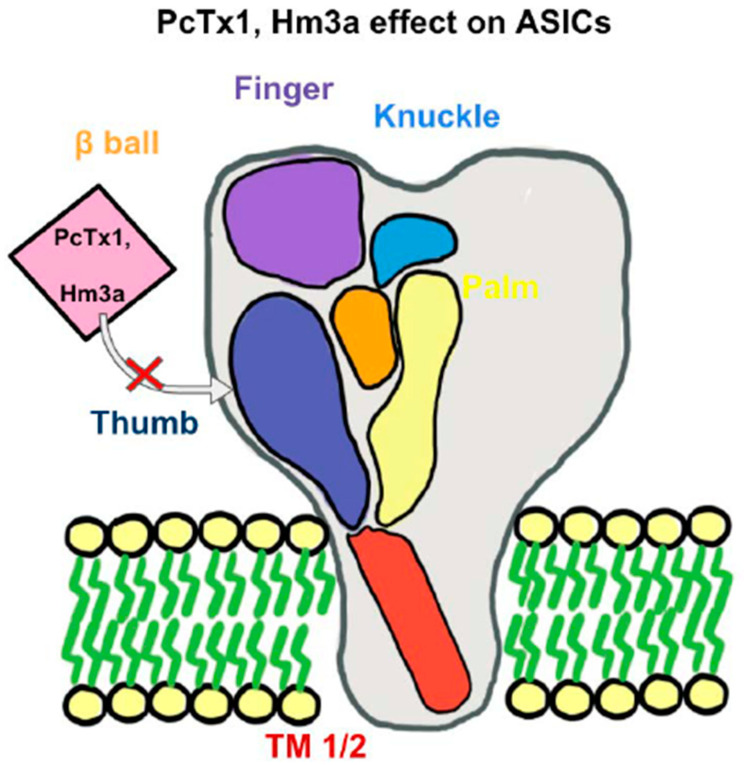
An Acid-Sensing Ion Channels (ASICs) subunit has a “clenched fist” conformation with six domains: wrist, finger, β-ball, thumb, knuckle, and palm domains. Combined, these subunits form a heterotrimeric or homotrimeric structure to help recognize extracellular ligands and regulate proton-gated currents. With the inhibition of the “thumb” component of an ASICs subunit, such as with PcTx1 or Hm3a, there will be an inhibition of certain ASICs channels. PcTx1 leads to the inhibition of ASIC1a, whereas Hm3a leads to the inhibition of both ASIC1a and ASIC1b and additionally higher levels of stability over a span of 48 h.

**Figure 2 ijms-22-04810-f002:**
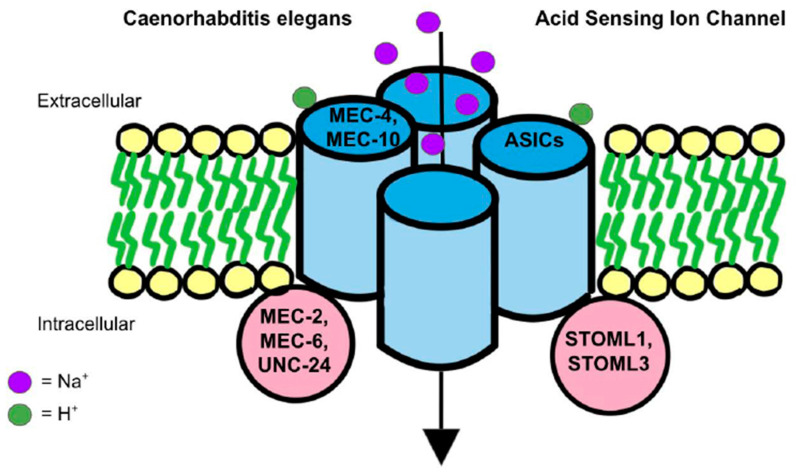
The mechanotransduction model found in Caenorhabditis elegans is the leading theory on the ASICs mechanosensation model in humans. On the left, as found in Caenorhabditis elegans, MEC-4 and MEC-10 are DEG/ENaC proteins that connect to touch receptor neurons and eventually act with a gating-spring mechanism to activate ASICs for mechanosensation. Similarly, intracellularly in nematodes, there are found to be MEC-2, MEC6, and UNC-24 accessory subunit proteins that loosely associate with ASICs to activate mechanosensation via a similar mechanism as well. Conversely, in mammals, ASICs are shown to contain STOML1 and STOML3 proteins which correlate with nematode accessory subunit proteins and help activate ASICs for mechanosensation as well.

**Table 1 ijms-22-04810-t001:** Mechanosensitive functions of ASICs.

Subtype of ASICs	Mechanosensitive Function
ASIC1	increases in sensitivity of mechanical forces in esophageal, colonic structures and gastric emptying [92,112]. has an important visceral mechanosensation process in urothelium and bladder compliance sensation [92,112]. is expressed in cutaneous Pacinian corpuscles and may serve as rapidly adapting low-threshold mechanoreceptors [114]. has an effect on primary hyperalgesia during inflammation, which is a local response at the area of injury [115]. contributes to peripheral vasoconstriction and vascular remodeling [117]. decreases mechanosensation in PNS with peripheral tissues including arteries, bone marrow, intestine, tongue and bladder [118,119,120]. ASIC1b is involved in pain sensation [24,25].
ASIC2	is linked to nociception and mechanosensation in the DRG [122,124]. is involved in mechanosensation in autonomic nervous system via the nodose ganglia [35]. is modulated by cardiac afferents to control blood pressure [124]. ASIC2a proteins have been found in Meissner, Merkel, penicillate, reticular, lanceolate, and hair follicle palisades in rat skin [110].
ASIC3	contributes to secondary hyperalgesia [115]. is associated with visceral colonic pain [48]. vasodilates small skeletal muscle arteries during muscle stress [135]. is heavily associated with nociception and proprioception, these functions are specifically channeled through Meissner corpuscles, and Merkel cells [110]. is heavily implicated in bladder physiology by providing sensory signaling during the filling of the bladder [49]. is involved in pain sensation in the bladder associated with inflammation [112]. contributes to neuronal mechanosensation that regulates changes in pH and motion found in lamprey models [139].

## Data Availability

Not applicable.

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
