# Peer review of "Acid-Sensing Ion Channels and Mechanosensation"

_ijms, 2021, doi:10.3390/ijms22094810_

Round 1

Reviewer 1 Report

The review manuscript by Ruan et al. highlights current data on the involvement of acid-sensing ion channels (ASICs) in various physiological and pathological processes of the central and peripheral nervous systems, such as learning, memory, synaptic plasticity, pain sensitivity, neurodegeneration, etc. The main emphasis is placed on the participation of ASIC channels in mechanosensation. This review presents all the basic information concerning the very interesting problem of the role of ASIC channels in the detection of various mechanical stimuli, which may expand the prospects of these channels as pharmacological targets.

As a basic recommendation: it would be useful to make a table that would summarize all the information on the participation of ASIC channels in mechanosensation. This would undoubtedly beautify and add visual representation to this review.

There are also a few comments and questions:

Line 30. Reference [2] does not quite fit the phrase "triggered by nonproton ligands during physiological pH levels", as it mainly talks about the action of protons. It would be nice to add a more relevant reference.

Line 33. ASICs belong to the degenerin/epithelial superfamily of channels.

Line 43. There is a missing reference to the article describing the calcium conductivity by the channel.

Line 46. In the above references, there is no information about the localization of channels in mitochondria and ER.

Line 72. Better to stick to one time; in lines 72 and 73 it is not entirely correct to use the past tense.

Line 80. It is not clear what the authors meant by this reference.

Line 89 -90. The sentence looks unfinished.

Line 141. For completeness, a description of the effect of neuropeptides such as dynorphins on neurodegeneration mediated by acting on ASIC1a could be added.

Line 196. The action of A-317567 is described in the present tense, although the effects of the previous ligands are described in the past tense.

Author Response

The review manuscript by Ruan et al. highlights current data on the involvement of acid-sensing ion channels (ASICs) in various physiological and pathological processes of the central and peripheral nervous systems, such as learning, memory, synaptic plasticity, pain sensitivity, neurodegeneration, etc.

The main emphasis is placed on the participation of ASIC channels in mechanosensation. This review presents all the basic information concerning the very interesting problem of the role of ASIC channels in the detection of various mechanical stimuli, which may expand the prospects of these channels as pharmacological targets.

Response: Thank you for your nice comments.

As a basic recommendation: it would be useful to make a table that would summarize all the information on the participation of ASIC channels in mechanosensation. This would undoubtedly beautify and add visual representation to this review.

Response: A new table on the participation of ASIC channels in mechanosensation has been added in our resubmission.

There are also a few comments and questions:

Line 30. Reference [2] does not quite fit the phrase "triggered by nonproton ligands during physiological pH levels", as it mainly talks about the action of protons. It would be nice to add a more relevant reference.

Response: A great point and thanks. A relevant reference was provided accordingly (Lines 434 to 435, reference 2).

Line 33. ASICs belong to the degenerin/epithelial superfamily of channels.

Response: Changed as suggested (Line 34).

Line 43. There is a missing reference to the article describing the calcium conductivity by the channel.

Response: A relevant reference from Dr. Welsh’s group published in 2004 has been cited (Lines 448 and 449, reference 10; Also see line 44).

Line 46. In the above references, there is no information about the localization of channels in mitochondria and ER.

Response: A new reference regarding the localization of channels in mitochondria has been cited (Lines 450 and 451, reference 11; Also see line 47).

Line 72. Better to stick to one time; in lines 72 and 73 it is not entirely correct to use the past tense.

Response: A great point and thanks. Changed as suggested (see lines 73 and 74).

Line 80. It is not clear what the authors meant by this reference.

Response: Thanks, and we cited a relevant reference (Lines 512 to 513, reference 39).

Line 89 -90. The sentence looks unfinished.

Response: Thanks, and we made changes (please see lines 92 to 93).

Line 141. For completeness, a description of the effect of neuropeptides such as dynorphins on neurodegeneration mediated by acting on ASIC1a could be added.

Response: New information regarding neuropeptides responsible for ischemic brain injury has been added (please see lines 147 to 148).

Line 196. The action of A-317567 is described in the present tense, although the effects of the previous ligands are described in the past tense.

Response: Changed as suggested (please see lines 203 to 204).

Reviewer 2 Report

Ruan and colleagues have presented a review article to deliver the concept of the roles of acid-sensing ion channels (ASICs) in mechanosensation. This is a timely issue for the readership in the field of mechanotransduction and sensory biology. However, the manuscript suffers from too many logical flaws and thus does not reach enough quality for publication.

  1. Basically, the structure of the manuscript should be further polished. The title is focusing on mechanotransduction, but half of content is not related to the topic.
  2. A review should provide a comprehensive viewpoint of the theme to be addressed and summarize important findings in this specific topic. In this aspect, a review should be comprehensive, tutorial, insightful, and providing specific viewpoints in the thematic topic. However, the manuscript looks like simply collecting information from 128 papers and put the information together without knowing what are the important and updated findings in this field, the knowledge gaps and challenges, the controversial debates, possible theories, etc.
  3. One of the major flaws is the accuracy of the content. Throughout the manuscript, the knowledge of the ASICs is often mis-interpreted, mis-understanding, misleading, cited with wrong and/or irrelevant references.
  4. I cannot list all of the wrong information (and wrong citation) in the manuscript, because there are too many. Beneath I just a few examples for the authors’ consideration.
  5. This subunit (ASIC2) exerts a protective effect against acidosis induced cell damage …. [39].
  6. Specifically, ASIC4 is seen to be the most active during osteoblastogenesis [53].
  7. It has been suggested that ASICs are involved in forming the acidic environment in the AD brain, although exact….[78,79].
  8. Chronic hyperalgesia in ASIC1b wildtype mice has been shown to last much longer than ASIC1b-KO mice [23].
  9. In a study carried out by Lee et al (2016), antihyperalgesesic medication….[25].
  10. Ref [47] [94][96] is not appropriate.
  11. STOML1 suppresses (but not activates) ASIC3 activation [102].
  12. Why syndecan-4 is relevant to ASICs?[103].
  13. ASIC3 has been shown to modulate vasodilation of small skeletal muscle arteries during muscle stress [125].
  14. In the introduction, ASIC2 involved in baroreception is mentioned, but this point is missing in the main chapter to describe the role of ASIC2 in mechanotransdction.
  15. It is not clear, why the role of ASIC3 in bladder is specifically focused.

Author Response

Ruan and colleagues have presented a review article to deliver the concept of the roles of acid-sensing ion channels (ASICs) in mechanosensation. This is a timely issue for the readership in the field of mechanotransduction and sensory biology. However, the manuscript suffers from too many logical flaws and thus does not reach enough quality for publication.

Basically, the structure of the manuscript should be further polished. The title is focusing on mechanotransduction, but half of content is not related to the topic.

Response: Thank you for your critical comments. We carefully and extensively polished our manuscript for resubmission. Also, we changed the tile of our manuscript in our resubmission. The revised title is “Acid-sensing ion channels and mechanosensation”.

A review should provide a comprehensive viewpoint of the theme to be addressed and summarize important findings in this specific topic. In this aspect, a review should be comprehensive, tutorial, insightful, and providing specific viewpoints in the thematic topic. However, the manuscript looks like simply collecting information from 128 papers and put the information together without knowing what are the important and updated findings in this field, the knowledge gaps and challenges, the controversial debates, possible theories, etc.

Response: Thank you for providing great tips regarding how to write a nice review article.

One of the major flaws is the accuracy of the content. Throughout the manuscript, the knowledge of the ASICs is often mis-interpreted, mis-understanding, misleading, cited with wrong and/or irrelevant references.

Response: We are so sorry for poor job in our original submission. We carefully checked each reference for our citation in the revised manuscript. We deleted all wrong/irrelevant references and provided relevant references in our resubmission. Also make sure not to mis-interpret, mis-understand and mislead the knowledge of the ASICs.

I cannot list all of the wrong information (and wrong citation) in the manuscript, because there are too many. Beneath I just a few examples for the authors’ consideration.

Response:  We made a detailed revision in our resubmission.

This subunit (ASIC2) exerts a protective effect against acidosis induced cell damage …. [39].

Response: We clarified this point. ASIC2 exerts its protection against acid-induced rat articular chondrocyte apoptosis through regulating ASIC1a expression and the intracellular Ca2+ levels and at least in part, suppressing p38 and ERK1/2 MAPK signaling pathways.

Specifically, ASIC4 is seen to be the most active during osteoblastogenesis [53].

Response: We clarified this issue. The revised description is as below: ASIC4, along with ASIC2 and ASIC3, are highly expressed during osteoblastogenesis in an acid environment [55]. Also, please see lines 104 to 105.

It has been suggested that ASICs are involved in forming the acidic environment in the AD brain, although exact….[78,79].

Response: We removed previous references and provided a newly relevant reference (#32). The changes are as below: It has been suggested that ASICs are involved in forming the acidic environment in the pathologic AD brain, although the exact underlying mechanism remains elusive [32]. Also, please see lines 161 to 162.

Chronic hyperalgesia in ASIC1b wildtype mice has been shown to last much longer than ASIC1b-KO mice [23].

Response: Thanks! Changed as suggested. The revision is as below: Transient and long-lasting mechanical hyperalgesia in ASIC1b wild-type (Asic1b+/+) mice has been shown to last much longer than in Asic1b−/− mice, though future research is needed to further delineate its mechanism [23]. Also, please see lines 174 to 176.

In a study carried out by Lee et al (2016), antihyperalgesesic medication….[25].

Response:  Thanks! The revision is as below: In a study carried out by Lee et al. (2018), antihyperalgesic medication at higher doses significantly reduced ASIC1b activity in rodents [25]. Also, please see lines 177 to 178.

Ref [47] [94][96] is not appropriate.

Response: Deleted as suggested and thanks.

STOML1 suppresses (but not activates) ASIC3 activation [102].

Response: We agree with the reviewer. The revision is as below: STOM is a protein that is likely attached to the C-terminus of MEC-4/MEC-10 to connect with the TM1 of ASIC3 to suppress the ion channel [102]. Also, please see lines 249 to 250.

Why syndecan-4 is relevant to ASICs? [103].

Response: We deleted the description regarding syndecan-4 related to ASICs.

ASIC3 has been shown to modulate vasodilation of small skeletal muscle arteries during muscle stress [125].

Response: The revisions are as below: ASIC3 has been activated during muscle ischemia. This activation helps induce the reflex baroreceptor response, which helps vasodilate the arteries that supply these skeletal muscles [135]. Also, please see lines 349 to 351.

In the introduction, ASIC2 involved in baroreception is mentioned, but this point is missing in the main chapter to describe the role of ASIC2 in mechanotransduction.

Response: New information regarding ASIC2 in mechanotransduction was added. See below: This function that ASIC2 has in regard to controlling blood pressure showcases the baroreceptive functions of ASIC2. For instance, in ASIC2 null mice, impaired baroreceptor reflex manifestations were shown, such as increased blood pressure and exaggerated sympathetic response, demonstrating the importance of ASIC2 in baroreception [35]. The underly mechanism of ASIC2 in the regulation of mechanosensitive properties might be due to membrane trafficking of ASIC2 proteins [126], which need to be examined by further study. Also, please see lines 323 to 329.

It is not clear, why the role of ASIC3 in bladder is specifically focused.

Response: It is well known that ASIC3 is involved in the pain modulation. ASIC3 also contributes to other functions, thus, we revised this issue not to put as much emphasis on the role that ASIC3 plays in the bladder. The revision is as below: Although ASIC3 largely contributes to pain modulation [46-48], ASIC3 also plays a role in the bladder. Also, please see lines 94 to 95.

Reviewer 3 Report

The manuscript is well written except for several parts.  I think authors can improve the review.

Lines184- 198  Discussion about molecules affecting ASICs. I think here authors should mention that variety of molecules were found and studied and give reference to any recent review. 

lines 261-262 rephrase the sentence " For example... " It is not very clear.

Author Response

The manuscript is well written except for several parts.  I think authors can improve the review.

Lines184-198  Discussion about molecules affecting ASICs. I think here authors should mention that variety of molecules were found and studied and give reference to any recent review. 

lines 261-262 rephrase the sentence " For example... " It is not very clear.

Response: Thanks for your nice comments!

Regarding discussion about molecules affecting ASICs in lines 184-198, we added recent reviews to this part. For example, we added references 83 and 84, which were newly published review articles regarding the pharmacology of the ASICs. Also, please see lines 606 to 608.

Regarding lines 261-262, we rephrased the sentence. The revisions are below: This stimulus differs from tissue to tissue. For example, in blood vessels, the laminar and oscillatory shear stress of blood stimulates the ion channel, while in peripheral sensory neurons, external physical forces stimulate the ion channel [106]. Also, please see lines 271 to 274.
